# Retinoic Acid-Differentiated Neuroblastoma SH-SY5Y Is an Accessible In Vitro Model to Study Native Human Acid-Sensing Ion Channels 1a (ASIC1a)

**DOI:** 10.3390/biology11020167

**Published:** 2022-01-20

**Authors:** Aleksandr P. Kalinovskii, Dmitry I. Osmakov, Sergey G. Koshelev, Kseniya I. Lubova, Yuliya V. Korolkova, Sergey A. Kozlov, Yaroslav A. Andreev

**Affiliations:** 1Shemyakin-Ovchinnikov Institute of Bioorganic Chemistry, Russian Academy of Sciences, ul. Miklukho-Maklaya 16/10, 117997 Moscow, Russia; kalinovskii.ap@gmail.com (A.P.K.); osmadim@gmail.com (D.I.O.); sknew@ya.ru (S.G.K.); lubova.ksenia@gmail.com (K.I.L.); july@mx.ibch.ru (Y.V.K.); serg@ibch.ru (S.A.K.); 2Institute of Molecular Medicine, Sechenov First Moscow State Medical University, 119991 Moscow, Russia

**Keywords:** acid-sensing ion channel (ASIC), neuroblastoma SH-SY5Y, neuronal differentiation, dopaminergic neurons, retinoic acid, drug development

## Abstract

**Simple Summary:**

Human neuroblastoma SH-SY5Y is used in neurobiology for studying various neuropathophysiological processes. In this study, we differentiated neuroblastoma cells into a neuronal-like phenotype with retinoic acid and studied if functional acid-sensing, transient receptor potential vanilloid-1 and ankyrin-1 ion channels were expressed in it. We found that homomeric acid-sensing ion channels 1a were expressed predominantly and yielded large ionic currents that can be modulated with different ligands. This channel plays important roles in synaptic plasticity, neurodegeneration, and pain perception. Thus, retinoic acid-treated neuroblastoma is a suitable model system for pharmacological testing on native human acid-sensing ion channels 1a. This approach can facilitate the development of new drugs for neuroprotection and pain management.

**Abstract:**

Human neuroblastoma SH-SY5Y is a prominent neurobiological tool used for studying neuropathophysiological processes. We investigated acid-sensing (ASIC) and transient receptor potential vanilloid-1 (TRPV1) and ankyrin-1 (TRPA1) ion channels present in untreated and differentiated neuroblastoma SH-SY5Y to propose a new means for their study in neuronal-like cells. Using a quantitative real-time PCR and a whole-cell patch-clamp technique, ion channel expression profiles, functionality, and the pharmacological actions of their ligands were characterized. A low-level expression of ASIC1a and ASIC2 was detected in untreated cells. The treatment with 10 μM of retinoic acid (RA) for 6 days resulted in neuronal differentiation that was accompanied by a remarkable increase in ASIC1a expression, while ASIC2 expression remained almost unaltered. In response to acid stimuli, differentiated cells showed prominent ASIC-like currents. Detailed kinetic and pharmacological characterization suggests that homomeric ASIC1a is a dominant isoform among the present ASIC channels. RA-treatment also reduced the expression of TRPV1 and TRPA1, and minor electrophysiological responses to their agonists were found in untreated cells. Neuroblastoma SH-SY5Y treated with RA can serve as a model system to study the effects of different ligands on native human ASIC1a in neuronal-like cells. This approach can improve the characterization of modulators for the development of new neuroprotective and analgesic drugs.

## 1. Introduction

Human neuroblastoma SH-SY5Y is a thrice-cloned subline of the cell line SK-N-SH that was derived from the bone marrow biopsy of a 4-year-old female patient. Since its establishment, this cell line has served as a tool for the study of various pathological conditions associated with neurodegenerative processes, such as Parkinson’s [1] and Alzheimer’s diseases [2], amyotrophic lateral sclerosis [3], ischemia [4], as well as neurotoxicity [5] and pathogenesis of viral infections [6,7]. This cell line is frequently preferred because of its human origin, catecholaminergic neuronal properties, and ease of maintenance [1,8]. The cells can be subjected to in vitro differentiation through several protocols to yield a phenotype that resembles dopaminergic human neurons. With retinoic acid (RA) treatment, neuroblastoma differentiation is accompanied by formation and extension of neuritic processes and the induction of neuron-specific enzymes, neurotransmitters, and neurotransmitter receptors [8]. Ion channel profiles in RA-treated neuroblastoma SH-SY5Y also alter, which is manifested in increased membrane excitability [9].

There is growing evidence that acid-sensing ion channels (ASICs) take a functional role in neuronal differentiation, where they modulate membrane excitability, maturation of dendrites and neurites, Ca^2+^ homeostasis, and dopamine secretion in dopaminergic neurons [10,11,12]. By virtue of being proton-gated Na^+^-channels, ASICs act as highly sensitive molecular sensors of extracellular pH change. There are six known isoform subunits that constitute trimeric functional ASICs, with ASIC1a being a predominant isoform found in the brain [13,14]. Generally, ASICs are abundant in neurons of the central and peripheral nervous systems where they are associated with synaptic plasticity, neuronal injury during brain ischemia, epileptic seizure termination, and axonal degeneration during multiple sclerosis as well as nociception and mechanosensation [15]. Their physiological roles make them potential therapeutic targets for pain management and neuroprotection. Much effort is invested in researching pharmacological ligands to these channels, because ASIC modulators can potentially be used in clinical practice [16,17]. The progress in this field may be facilitated by the availability of new screening model systems whose properties mimic human neurons.

In the present research, we examined RA-treated human neuroblastoma SH-SY5Y in comparison with its untreated counterpart as a prospective in vitro model system to study native acid-sensing, transient receptor potential vanilloid-1 (TRPV1) and ankyrin-1 (TRPA1) ion channels. In electrophysiological measurements, we did not observe any response to selective TRPV1 and TRPA1 agonists in RA-treated neuroblastoma cells. However, RA-treated cells showed large ASIC-like currents in response to acid stimuli that were dramatically greater in amplitude than the currents of the untreated cells. Real-time PCR quantification revealed that with ongoing differentiation ASIC1a emerged as a prevalent isoform of ASICs, while ASIC2 expression was almost unaltered, and ASIC3 expression was lacking in both untreated and differentiated cells. TRPV1 and TRPA1 channel transcripts significantly diminished with RA treatment and their currents were detected only on a limited fraction of undifferentiated cells. We provide a detailed kinetic and ligand-assisted characterization of ASICs present in RA-treated SH-SY5Y, and suggest that this in vitro model may be useful for further ASIC1a research.

## 2. Materials and Methods

### 2.1. Reagents and Ligands

Retinoic acid, capsaicin, ASP7663 and amiloride were obtained from Sigma-Aldrich (St. Louis, MO, USA). Sevanol was obtained by chemical synthesis [18]. Peptides APETx2 and mambalgin-2 were produced with heterologous expression in *Escherichia coli* [17,19]. Lyophilized compounds were dissolved in fresh buffers directly prior to experiments in extracellular solution.

### 2.2. Cell Culture

Experiments were conducted on human neuroblastoma SH-SY5Y cells (CLS Cat# 300154/p822_SH-SY5Y, RRID:CVCL_0019) obtained from ATCC, USA. SH-SY5Y cells were cultured in Basic Growth Media (DMEM/F12 (1:1) medium (Thermo Fisher Scientific, Waltham, MA, USA) supplemented with 2 mM L-glutamine, 10% fetal bovine serum (HyClone, Logan, UT, USA),100 U/mL penicillin, 100 μg/mL streptomycin) at 37 °C in a humidified atmosphere with 5% of CO_2_. Cells were split 1:3 or 1:4 every 5–7 days, the number of passages did not exceed 10–15.

For SH-SY5Y cell differentiation, a one-step protocol was used: on day 0 (d0), cells were seeded in Basic Growth Media onto a fresh cell culture dish or cover slips so that the confluence level was about 40–50%; on day 1 (d1), the medium was exchanged to the differentiation medium (DMEM/F12 (1:1)) with 2 mM L-glutamine, 1% fetal bovine serum, 100 U/mL penicillin, 100 μg/mL streptomycin, 10 μM RA); cells were protected from light. Patch-clamp experiments were performed on days 6 (d6)–8 (d8), and cells for real-time PCR were harvested on d6.

### 2.3. RNA Extraction and cDNA Synthesis

Total RNA was extracted from defrosted and homogenized SH-SY5Y cells with TRIzol™ Reagent (Life Technologies, Carlsbad, CA, USA) and the following chloroform phase separation and ethanol precipitation at −20 °C. Washed RNA was dissolved in Steril RNAse-free water (Thermo Fisher Scientific, USA). Final RNA concentration was measured using a spectrophotometer. To prevent RNA aggregation, total RNA samples were heated at 65 °C for 1–2 min before cDNA synthesis. First-strand cDNA synthesis was performed using Mint Kit (Evrogen, Moscow, Russia) according to the manufacturer’s protocol (1.5 μg RNA for each reaction). 

### 2.4. Quantitative PCR

Real-time PCR was performed using SsoAdvanced Universal SYBR Green Supermix (Bio-Rad, Hercules, CA, USA) in a 7500 real-time PCR system (Applied Biosystems, Waltham, MA, USA). We used the following primers:

ASIC1a: fwd AGCGGCTGTCTCTGAAGC, rev AGCTCCCCAGCATGATACAG (specific for ASIC1a transcript (transcript variant 2) [20], does not recognize mRNA of ASIC1b isoform (transcript variant 3); ASIC2: fwd TTGGCGGAAAAGGACAGC, rev AGCAACTTCATACGCCTTCTTCTG (detects all ASIC2 transcripts); ASIC3: fwd CCGCGGAGAACGTGCTG, rev CTTGTCTCGGAACACCTCAC; TRPV1: fwd CGCTGATTGAAGACGGGAAG, rev CAGGAGGATGTAGGTGAGAATTAC; TRPA1: fwd GGAACACTGCACTTCACTTTG, rev CATGCATTCAGGGAGGTATTC; β-actin: fwd CCACGAAACTACCTTCAACTCC, rev TCGTCATACTCCTGCTTGCTGATCC.

The quantity of transcripts was normalized to that of β-actin. Samples without cDNA served as negative controls for each gene. The real-time PCR protocol was as follows: long denaturation phase (95 °C, 10 min), 35 cycles of denaturation (95 °C, 15 s), annealing (60 °C, 10 s), and elongation (72 °C, 30 s). The resulting PCR products created single bands of appropriate sizes in agarose gel electrophoresis. Each reaction was performed in at least six technical replicates. Data were analyzed using 7500 real-time PCR system Software (comparative ΔΔC_T_ method). ΔC_T_ method: ΔC_T_ = (C_T_ gene of interest—C_T_ β-actin); 2 ^−ΔC_T_^ is the measure of the mRNA expression level in the sample normalized to the housekeeping gene β-actin. ΔΔC_T_ method: ΔΔC_T_ = [(C_T_ gene of interest- C_T_ β-actin) cDNA of RA-treated neuroblastoma cells]—[(Ct gene of interest-Ct internal control) cDNA of neuroblastoma cells]; 2 ^−ΔΔC_T_^ = Relative Quantification (RQ) value that is the measure of the mRNA expression in the sample normalized to the control sample.

### 2.5. Electrophysiology

Borosilicate glass pipettes (outside diameter 1.5 mm, inside diameter 0.86 mm) were pulled with a P-1000 flaming/brown micropipette puller (Sutter instrument, Novato, CA, USA). The pipette resistance ranged from 7 to 10 MΩ when filled with the solution that contained 5 mM MgCl_2_, 100 mM CsF, 10 mM 4-(2-hydroxyethyl)-1-piperazineethanesulfonic acid (HEPES) (pH 7.2 adjusted with 1 M KOH, osmolarity adjusted with 1M sucrose). Extracellular solution contained 2 mM CaCl_2_, 1 mM MgCl_2_, 125 mM NaCl, 4 mM KCl, 10 mM glucose, 5 mM HEPES (pH 7.4 adjusted with 1 M NaOH). Activating solutions of the same content were titrated by 1 M NaOH to the required pH and supplemented with 10 mM HEPES (pH 7.15–7.55), or 3-(N-morpholino)propanesulfonic acid (MOPS) (pH 6.15–7.15), or 2-(N-morpholino)ethanesulfonic acid (MES) (pH < 6.15). The difference in osmolarity between pipette and extracellular solutions was 10–15 mOsm/L.

Whole-cell patch-clamp recordings were performed with EPC-800 amplifier (HEKA, Lambrecht, Germany), InstruTECH LIH 8 + 8 data acquisition system (HEKA) and PatchMaster v2x90.5 software (HEKA). The membrane potential was clamped at −90 mV. Rapid application of extracellular solutions was achieved with an in-house two-barrel perfusion system driven by SF-77B Perfusion Fast Step (Warner instruments, Holliston, MA, USA). The interval between applications was no less than 1 min to achieve complete recovery from desensitization. For TRPA1 measurements, membrane potential was clamped at −40 mV. TRPA1 inward/outward currents were elicited with a repeated voltage protocol (Appendix A, Figure A1) in the absence and in the presence of agonist. Experiments were conducted at room temperature (22–24 °C). Data were filtered at 3 kHz, digitized at 0.5–1 kHz and analyzed using PatchMaster and OriginPro 8.6 (OriginLab, Northampton, MA, USA).

### 2.6. Data and Statistical Analysis

Data were fitted with the logistic equation: F (x) = A/(1 + (x/[H^+^]_50_)^n_H_), where F (x) is the current amplitude at the proton concentration x; [H^+^]_50_ is the concentration of protons at which half-maximal current was obtained; n_H_ is the Hill coefficient; A is the maximal current amplitude (I_max_). I_max_ was calculated for each cell by individual fitting and taken for data normalization. The rate of current decay was fitted using a single exponential equation F (t) = A *exp (−t/τ_des_) + A_0_, where F (t) is the current amplitude at the time t; τ_des_ is the τ of desensitization; A_0_ is the baseline current amplitude. All data are presented as mean ± SEM (standard error of mean). To normalize data on the effect of ASIC inhibitors, all the control currents were polled in one data set and the experimental currents were normalized to the average of the control currents. The significance of the data differences was determined with a one-way analysis of variance (ANOVA), followed by Tukey’s test. For real time PCR data, difference significance was determined using the non-parametric Kruskal–Wallis test. *p* < 0.05 was considered as significant.

## 3. Results

### 3.1. Quantitative mRNA Difference in Untreated and RA-Treated SH-SY5Y Cells

To clarify the possibility of using SH-SY5Y cells in the study of acid-activated ion channels, we determined the levels of mRNA transcripts in untreated cells and cells treated with RA for ASIC1a, ASIC2, ASIC3, TRPV1, and TRPA1 ion channels. Quantitative real-time PCR experiments showed that ASIC1a seems to be the main ASIC isoform expressed in differentiated SH-SY5Y cells: the number of mRNA transcripts was significantly higher on day 6 of RA-treatment compared with the control (untreated cells) (Figure 1A,B; Appendix A, Figure A2). ASIC2 isoform was also present in SH-SY5Y cells, though the expression levels of ASIC2 did not change significantly during differentiation, and the number of ASIC2 transcripts was by an order of magnitude smaller than ASIC1a (Figure 1A). Finally, ASIC3 showed no detectable expression up to 40 cycles in either untreated or differentiated SH-SY5Y cells. We also detected changes in TRPV1 expression: the number of TRPV1 transcripts decreased by d6 (Figure 1A,B). The mRNA level of TRPA1 was extremely low, suggesting that TRPA1 would be difficult to detect in electrophysiological measurements.

### 3.2. Comparison of Acid-Induced Currents in Untreated and RA-Treated SH-SY5Y Cells

Retinoic acid-treated cells notably differed in morphological traits from untreated cells, which was in good agreement with the previously reported descriptions [21,22,23]. The 6-day incubation with RA resulted in a decreased clustering and a raised number of smaller neuroblast-like cells (N-cells), which acquired a fusiform shape and formed neuritic processes (Figure 2).

N-cells (some representatives are marked with black arrows in Figure 2) were further selected for electrophysiological experiments, since epithelial-like S-cells, which had flat and angular shape, gave no responses to the acidic stimuli (Appendix A, Figure A3).

The ability of untreated and RA-treated SH-SY5Y cells to respond to proton stimuli was investigated using a whole-cell patch-clamp technique. Inward currents were recorded with a rapid change of the extracellular pH from 7.4 to 6.0. In response to acidic stimuli, untreated cells showed rapidly developing ASIC-like currents with the mean amplitude of 141.3 ± 32.4 pA (n = 14), whereas RA-treated SH-SY5Y cells responded to acidic stimuli with the mean amplitude of 485.5 ± 37.4 pA (n = 25) (n = 25) (Figure 3A,B). Thus, RA-induced differentiation of SH-SY5Y cells remarkably enhanced their responsiveness to protons resulting in currents that were 3.5 times greater in amplitude than those of untreated cells. Acid-induced currents of untreated and RA-treated SH-SY5Y cells fitted well with a one-exponential function (see Materials and Methods) and showed similar non-significantly different desensitization time constants (τ_des_) with values of 818.8 ± 40.7 ms for untreated (n = 11) and 750.6 ± 35.6 ms for RA-treated cells (n = 28) (Figure 3C).

We tested untreated and RA-treated SH-SY5Y cells for their ability to respond to TRPV1 and TRPA1 selective agonists to verify the presence or absence of these channels. We used capsaicin for TRPV1 activation (EC50 = 0.11–1.11 µM for human TRPV1 [24,25]) and ASP7663 for TRPA1 activation (EC_50_ = 0.51 µM for human TRPA1 [26]). Application of 1 and 4 µM of capsaicin (n = 6 and 10, respectively) and 0.7 µM of ASP7663 (n = 8) did not induce detectable currents in RA-treated cells (Figure 4A,B, right panel). 

In untreated SH-SY5Y cells, none of the cells responded to 1 µM capsaicin stimulus (n = 7), and only 20% of cells responded to 4 µM capsaicin stimulus (n = 10) and 10% to 0.7 µM ASP7663 (n = 10) (Figure 4A,B, left panel). These data are consistent with the results of quantitative real-time PCR experiments, which showed that mRNA abundance of TRPV1 and TRPA1 was significantly lower in RA-treated cells (Figure 1). The consecutive application of 4 µM capsaicin and acid stimulus did not give any response to TRPV1 agonist and did not diminish acid-induced currents with any significance compared with the control (pre-application of pH 7.4). This suggests that TRPV1 rather does not contribute to acid-induced currents on untreated cells because the distribution of functional channels is very scarce (Appendix A, Figure A4).

### 3.3. Characterization of ASIC Currents in RA-Treated SH-SY5Y Cells

The rapidly developing and rapidly desensitizing acid-induced currents are normally mediated by ASIC channels. Human TRPV1 and TRPA1 can respond to acid stimuli but there were no responses to selective agonists of these channels in RA-treated cells. Moreover, human TRPA1 mediates slow inward currents in response to pH drop, and because of their distinct kinetic properties, they can be easily isolated from ASIC-like currents [27]. We also did not detect any response to capsaicin (TRPV1 agonist) in pH-responsive RA-treated neuroblastoma cells (Figure 4C). According to the results of RT-PCR, both ASIC1a and ASIC2 are present in RA-treated cells. To establish which isoform makes the greatest contribution to the cell responsiveness to acid stimuli, desensitization time constants (τ_des_), pH of half-maximum activation (pH_50_), and pH of half-maximum steady-state desensitization (pHSSD_50_) were determined. The τ_des_ values were measured at activating impulses of pH 6.0 and 5.0. With the activation impulse of 6.0, the τ_des_ value was 750.6 ± 35.6 ms (n = 28), while at pH 5.0, the τ_des_ value decreased significantly to 490.2 ± 49.6 ms (n = 15) (Figure 5D).

τ_des_ of homomeric ASIC1a channels was previously reported to be in the range from 1 to 2 s [12,28,29], while τ_des_ < 1 s was reported to be inherent to heteromeric ASIC1a/ASIC2a channels [28]. However, for heteromeric ASIC1a/ASIC2a channels, it was also shown that with greater acid stimulus τ_des_ significantly increases and approaches a value close to 1 s. In RA-treated cells, τ_des_ significantly decreased with stronger acid stimulus and matched perfectly to the value for homomeric ASIC1a channels [28].

The pH dependence of activation, measured from 10 activating pH values, gave the pH_50_ and Hill’s coefficient (cooperativity factor, n_H_) values equal to 6.65 ± 0.01 and 4.6 ± 0.4, respectively (n = 12) (Figure 5A,C). The pH dependence of steady-state desensitization, i.e., such conditioning pH values that cause transient current desensitization without apparent activation, was measured from 6 conditioning pH values; pHSSD_50_ and n_H_ were 7.19 ± 0.01 and 8.4 ± 0.6, respectively (n = 7) (Figure 5B,C). The obtained pH_50_ and pHSSD_50_, as well as their n_H_ values, are also in excellent agreement with those obtained previously for homomeric ASIC1a channels [12,28,30,31], which convincingly confirms that homomeric ASIC1a channels principally determine proton sensitivity in RA-treated SH-SY5Y cells.

### 3.4. Pharmacological Characteristics of ASIC1a in RA-Treated SH-SY5Y Cells

We performed pharmacological characterization of acid-induced currents in RA-treated SH-SY5Y cells using four ASIC channel ligands: low-molecular compounds amiloride and sevanol, and peptides mambalgin-2 (Mamb-2) and APETx2. Ligands were applied only after a stable response to pH 6.0-induced activation was achieved. Amiloride, which finds a medical application as a K^+^-sparing diuretic agent, is a non-specific blocker of ASIC channels [32]. Pre-application of 500 µM amiloride resulted in the inhibition of the pH 6.0-induced current by 39 ± 5% (n = 8) (Figure 6A,F,J).

Sevanol, which inhibits ASIC1a and ASIC3 isoforms and does not act on ASIC2a [18,33], at the concentration of 300 μM demonstrated a stronger effect than amiloride, suppressing acid-induced currents of RA-treated SH-SY5Y cells by 80.9 ± 5.5% (n = 9) (Figure 6B,G,J). Peptides Mamb-2 and APETx2 are known as selective ligands of the ASIC1 and ASIC3 isoforms, respectively [34,35]. A 12-s pre-application of Mamb-2 (3 µM) inhibited pH 6.0-induced currents by 59.1 ± 4.9% (n = 9) (Figure 6C,H,J). APETx2 (1 µM) under the same conditions caused an insignificant reduction in the response to an acid stimulus by only 4.5 ± 15.2% (n = 8); however, a raise in concentration to 10 µM led to an increase in the effect to 23.8 ± 11.6%, although it was also statistically non-significant (n = 12) (Figure 6D,E,I,J). Therefore, acid-induced currents of RA-treated SH-SY5Y cells are effectively inhibited only by substances that suppress the functioning of the ASIC1a isoform.

We also conducted a comparative test of the ligands’ action on SH-SY5Y cells (Appendix A, Figure A5). Sevanol, at the concentration of 300 μM, suppressed acid-induced currents of untreated cells by 38.2 ± 3.3% (n = 5), which is two times less than the effect on currents from RA-treated cells. Since sevanol does not act on the ASIC2a isoform [33], the effect on heteromeric ASIC1a/ASIC2 has not yet been reported, and pH 6.0 does not activate homomeric ASIC2a channels [28], we suppose that the currents from untreated cells are partially mediated by channels containing ASIC2 subunits in heteromeric form. Peptide Mamb-2 at the concentration of 3 µM inhibited pH 6.0-induced currents by 49.2 ± 1.7% (n = 5): significantly lower than the effect on the currents of RA-treated cells. Mamb-2 was previously reported to act less effectively on heteromeric ASIC1a/ASIC2a (IC_50_ = 246 nM) compared with homomeric ASIC1a (IC_50_ = 55 nM) [34,36]. Thus, the negative shift in Mamb-2 efficiency possibly confirms a greater contribution of ASIC2 subunits in mediating currents of untreated SH-SY5Y cells.

## 4. Discussion

In vitro cell lines of a neuronal nature have significantly advanced progress in neurobiology. Experimental models that are employed to study the nervous system functioning in vitro include primary neurons of rodents, rodent and human neuroblastoma cell lines, stem cells and immortalized neuronal lines. Originally established in 1970, human neuroblastoma SH-SY5Y is a malignant cell line that has become particularly prominent for neurobiological research. Although in a plethora of research intact SH-SY5Y is employed [1], the cells can undergo in vitro differentiation resulting in a dopaminergic-like phenotype. The reported differentiation methods include the treatment with solely retinoic acid (RA) or dibutyryl cyclic AMP [37], or sequential treatment with RA and 12-O-tetradecanoylphorbol-13-acetate [38], brain-derived neurotrophic factor [39], or cholesterol [40]. The advantages of differentiated SH-SY5Y include cell cycle synchronization in G1 phase during differentiation, relative ease of maintenance, low cost, compared to primary neurons, as well as escape from ethical complications which occur when using human neuronal cultures.

TRPV1 and TRPA1 play important roles in inflammatory and neurodegenerative conditions [41,42]. According to our findings, these channels are lacking in RA-treated neuroblastoma SH-SY5Y cells and are rather rare in the untreated cell line. Therefore, untreated and RA-treated neuroblastoma SH-SY5Y cells are not a suitable model for the study of TRPV1 and TRPA1. Probably, other protocols of differentiation could lead to cells expressing these channels.

The important role of ASICs in neuronal functioning is recognized due to their involvement in neurodegenerative, adaptive, and cognitive processes [43,44,45,46]. In this study, we investigated ASICs on a model of RA-treated neuroblastoma SH-SY5Y. This model represents a simple and effective means to study native ASIC1a in neuronal-like cells. It was previously reported that ASIC channels are present in undifferentiated SH-SY5Y cell line, but the pH of half-maximal activation (pH_50_) equal to 6.01 [47] only suggested a possible ASIC1a/ASIC2a/ASIC3 subunit composition of the channels [28]. Together with relatively small amplitudes (from 137.71 pA at pH 6.0 to 270.37 pA at pH 5.0) [47], the undifferentiated cell line did not look attractive as a model for ASIC channel study. In RA-treated cells, we demonstrated that the peak current amplitude increased more than threefold. Moreover, the data of quantitative real-time PCR and the proton affinity parameters point to the prevalence of homomeric ASIC1a channels. Indeed, pH_50_ and pHSSD_50_, equal to 6.65 and 7.19, respectively, as well as desensitization time constant (τ_des_), equal to 0.49 s at the pH 5.0 stimulus, perfectly correspond to the known values for homomeric ASIC1a [12,28,30]. The established dominance of ASIC1a is consistent with previous reports of its involvement in neurodifferentiation where this channel apparently modulates maturation of dendrites and neurites and dopamine secretion in dopaminergic neurons [10,11,12]. Previously, it was shown that the direction of axon growth cones is guided by the joint switch-like action of calcineurin and Ca^2+^/calmodulin-dependent kinase type II (CaMKII) [48]. The latter was also shown to affect the length of cytoskeletal proteins such as F-actin [49]. Homomeric ASIC1a are permeable to Ca^2+^, and their activation was reported to be functionally coupled to the activation of CaMKII [50]; therefore, the role of ASIC1a in neurodifferentiation could possibly be mediated by the CaMKII signaling pathway. Fine details of ASIC1a involvement in neurodifferentiation expect further elucidation.

With the assistance of four ligands, we performed the pharmacological characterization of the ASIC channels present in the RA-treated cells. The effectiveness of the ASIC1a ligands also confirms the eligibility of this model for characterizing homomeric ASIC1a channels. Sevanol and mambalgin-2, which act on the ASIC1a isoform, produced the most significant effect on acid-induced currents. Amiloride displayed rather weak inhibition, and the ASIC3-specific peptide APETx2 did not have a significant effect at all, which is consistent with the lack of ASIC3 mRNA in quantitative real-time PCR. The fact that 10 μM APETx2 still led to the current decrease by 25–30% is in good agreement with the previous report that at this high concentration, APETx2 produces nonspecific modulation of other ASICs, including a slight ASIC1a inhibiting effect [51]. The test of ASIC1a ligands on untreated SH-SY5Y showed that their effects are lower, which possibly indicates to a greater fraction of ASIC2 that are responsible for mediating acid-induced currents of untreated cells.

The ligands often do not act with the same efficiency on human ASICs and rodent orthologues [52,53,54]; therefore, for effective drug development it might be crucial to recruit test systems based on human cells. In fact, few results from animal experimentations can be reliably transferred to humans [55,56]. Preliminary testing on heterologously expressed human or rodent channels in cell systems can overestimate or underestimate the desired therapeutic effect and predetermine a possible failure of clinical trials. 

## 5. Conclusions

Ion channels sensitive to acid (ASICs, TRPV1, TRPA1) play important roles in inflammatory, neurodegenerative, adaptive, and cognitive processes. New neuronal-like cell model systems are an attractive means for their pharmacological and physiological study. The present work demonstrates that retinoic acid-treated SH-SY5Y cells are an effective model for the study of native human ASIC1a channels. We showed that the neuronal differentiation was accompanied by a significant rise in acid-induced currents. We proceeded to show that they were prevalently mediated by homomeric ASIC1a channels, unlike acid-induced currents from untreated cells that were mediated by a heterogeneous mixture of channels. The test system based on RA-differentiated neuroblastoma SH-SY5Y, which bears intrinsic human-specific properties, may serve as an instrument for attempts to make accurate assessments of promising compounds acting on human ASIC1a.

## Figures and Tables

**Figure 1 biology-11-00167-f001:**
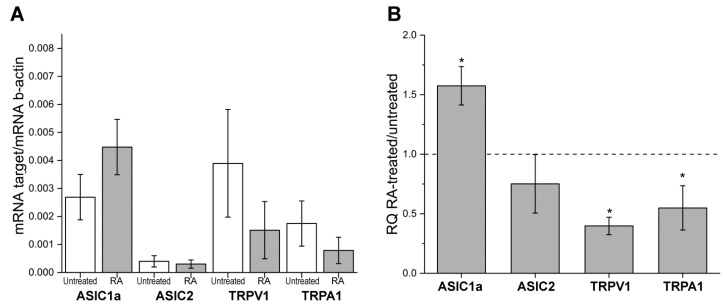
The relative levels of ASIC1a, ASIC2, TRPV1, and TRPA1 mRNA transcripts at d6. (**A**) mRNA levels of ion channels in control (untreated) and differentiated (with 10 µM RA) SH-SY5Y cells normalized to the housekeeping gene β-actin (ΔC_T_ method). (**B**) mRNA levels of ion channels in RA-treated SH-SY5Y cells normalized to the housekeeping gene β-actin and the corresponding mRNA of untreated SH-SY5Y (RQ, ΔΔC_T_ method). There was no detectable expression of ASIC3 in either untreated or differentiated SH-SY5Y cells. Data shown as mean ± SEM. Data are from 3 independent experiments, with 6 technical replications each. Statistical analysis was performed using the Kruskal–Wallis ANOVA test, *—*p* < 0.05.

**Figure 2 biology-11-00167-f002:**
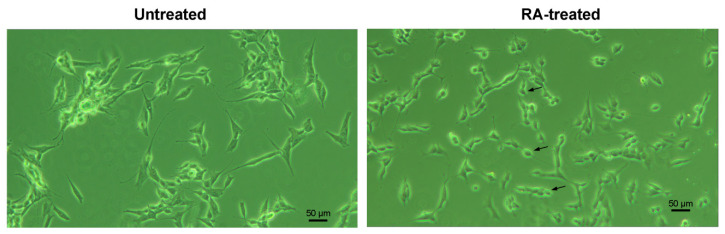
Morphological changes in SH-SY5Y cells after the treatment with retinoic acid (RA). Observations were made under a light microscope on the 6th day after incubation in 1% serum medium containing 10 µM RA. Arrows indicate cells selected for the electrophysiological testing. Scale bar, 50 μm.

**Figure 3 biology-11-00167-f003:**
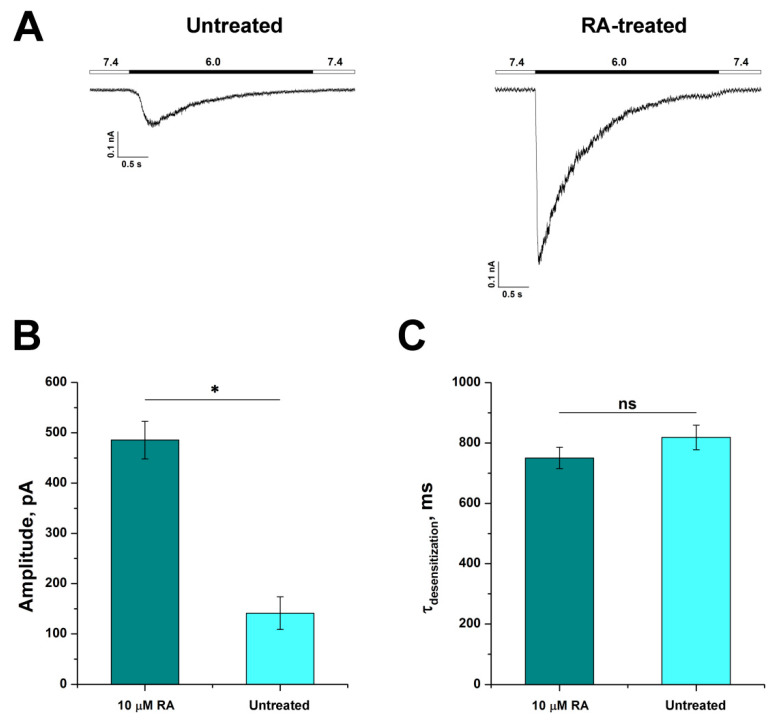
Comparison of acid-induced currents of untreated and 6 days RA-treated (6d) SH-SY5Y cells. (**A**) Whole-cell current traces recorded from untreated and 6d RA-treated cells in response to the pH 6.0 stimulus. The kinetic variations of ASIC responses to the pH 6.0 stimulus: (**B**) variation of the peak amplitude value; (**C**) variation of the desensitization time constant (τ_des_). Each bar is presented as mean ± SEM of 11–28 measurements. Statistical analysis was performed with ANOVA followed by Tukey’s test. *—*p* < 0.05 is significantly different; ns, non-significant difference.

**Figure 4 biology-11-00167-f004:**
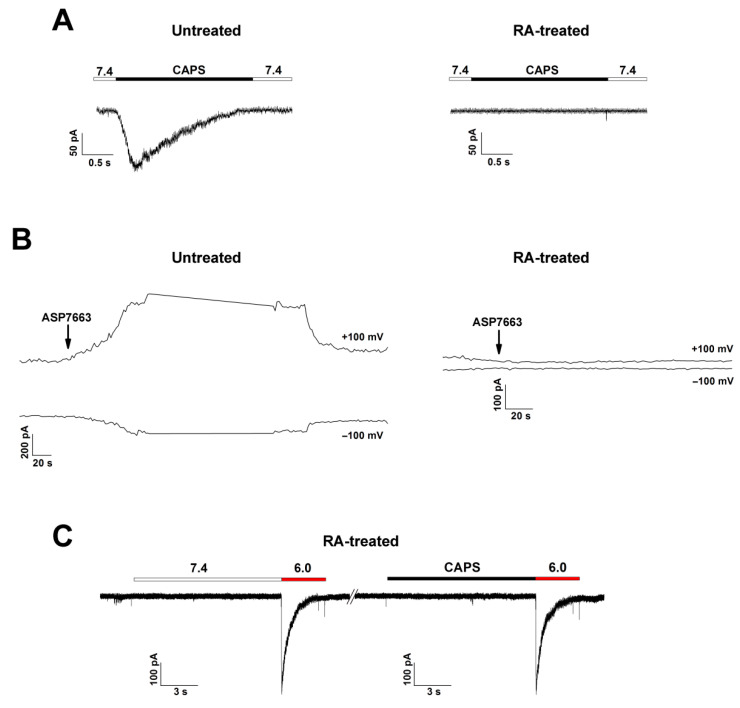
Detection of TRPV1- and TRPA1-mediated currents in untreated and RA-treated SH-SY5Y cells. (**A**) Whole-cell current traces recorded from RA-treated and untreated cells in response to 4 µM capsaicin stimulus applied at pH 7.4. (**B**) Representative current traces evoked by 0.7 µM ASP7663 and recorded at +100 and –100 mV. (**C**) Representative current traces evoked by pH 6.0 after application of both control buffer (pH 7.4) (left panel) and 4 µM capsaicin (right panel) recorded from the same cell of RA-treated neuroblastoma.

**Figure 5 biology-11-00167-f005:**
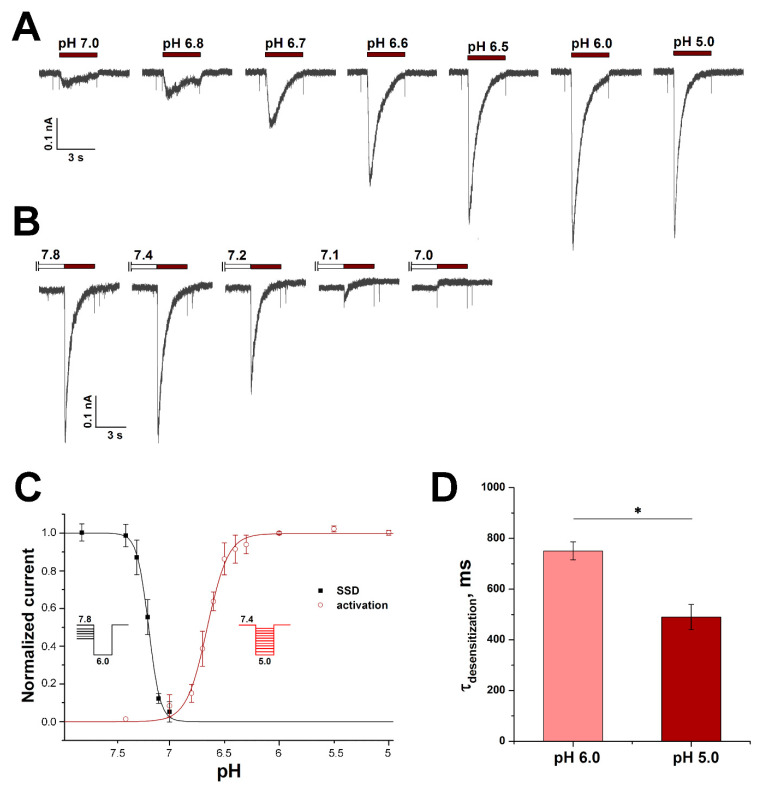
pH dependence of ASIC1a activation and desensitization in RA-treated SH-SY5Y cells. (**A**) Representative whole-cell ASIC current traces elicited by rapid pH change from 7.4 to the values in the range from 7.0 to 5.0 for 3 s. (**B**) Representative current traces elicited by application of pH 6.0 for 3 s from conditioning pH values in the range from 7.8 to 7.0 pre-applied for 12 s. Whole-cell patch-clamp currents were measured at the holding potential of −90 mV. (**C**) pH dependence of ASIC steady-state desensitization (SSD) (black line) and activation (red line). Normalized currents are plotted as a function of the conditioning and stimulation pH for the SSD and activation curves, respectively. (**D**) Comparison of τ_des_ values of currents elicited by pH 6.0 (n = 28) and 5.0 (n = 15) stimuli. Each bar is presented as mean ± SEM. Statistical analysis was performed with ANOVA followed by Tukey’s test. *—*p* < 0.05 is significantly different.

**Figure 6 biology-11-00167-f006:**
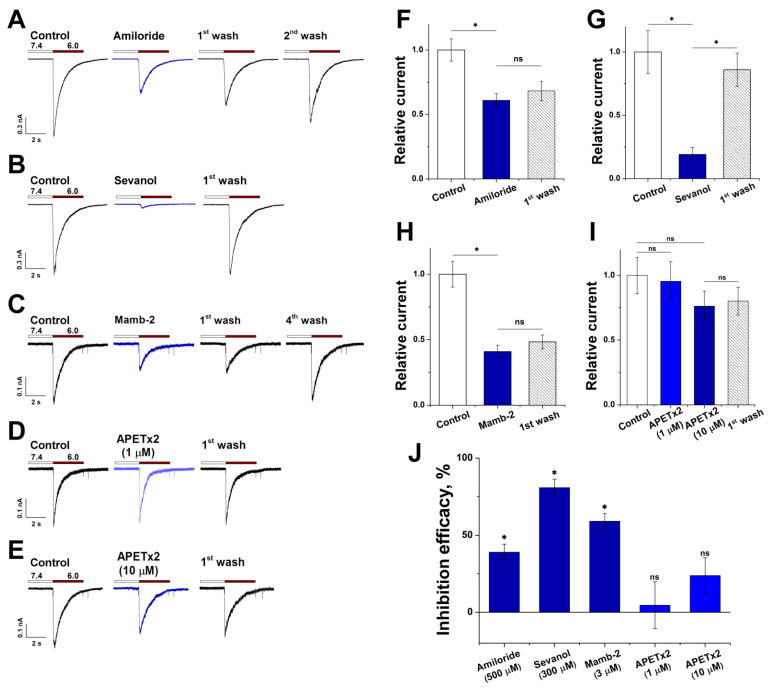
Pharmacological characteristics of ASIC-mediated currents in RA-treated SH-SY5Y cells. Representative current traces obtained with the pH drop from 7.4 to 6.0 with (blue) or without (black) 12 s pre-incubation with 500 µM amiloride (**A**), 300 µM sevanol (**B**), 3 µM mambalgin-2 (Mamb-2) (**C**), 1 and 10 µM APETx2 (**D**,**E**). Bar charts of peak amplitudes of pH 6.0-induced currents after a 12 s pre-incubation with 500 µM amiloride (n = 8) (**F**), 300 µM sevanol (n = 9) (**G**), 3 µM Mamb-2 (n = 9) (**H**), 1 and 10 µM APETx2 (n = 8 and n = 12, respectively) (**I**). White and grey bars are controls before and after ligand action, respectively. (**J**) The summary bar chart of the activity of all tested ASIC ligands. Statistical analysis was performed with ANOVA followed by Tukey’s test. *, *p* < 0.05 vs control group; ns, non-significant in comparison control group.

## Data Availability

The data that support the findings of this study are available from the corresponding author upon reasonable request. Some data may not be made available because of privacy or ethical restrictions.

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
