# Peer review of "Retinoic Acid-Differentiated Neuroblastoma SH-SY5Y Is an Accessible In Vitro Model to Study Native Human Acid-Sensing Ion Channels 1a (ASIC1a)"

_biology, 2022, doi:10.3390/biology11020167_

Round 1
Reviewer 1 Report
In this manuscript, Kalinovskii et al. showed that by treating SH-SY5Y cell line with RA, the expression of ASIC1a in cells would increase exponentially. The methods are well designed and the manuscript reads well; however there are some issues that the authors need to address. Please see my comments.
Major comments:
- The time course of activation of ASIC1a in the control and treated cells were different. If the same channel would be the responsible in both group; why there is the time course delay? Therefore, I believe to confirm this, the authors needs to repeat the experiments mentioned in figure 6 on the untreated cells. This way they can claim that the same or different channel is responsible for the findings observed in figure 4a.
- The authors mentioned that ASIC1a is involved in CAMKII maturation; it would be good that the authors compare the mRNA expression of CAMKII in the treated and untreated cells.
Minor comments:
- line 22: remove "." after "acid-sensing ion channels.".
- In many places the authors have not defined the acronyms first such as in line 25, ASIC and TRP; please check the whole manuscript.
- Lines 44-47, the sentence "Human neuroblastoma..." does not read well; please revise.
- Line 68, please remove "." after "clinical practice."
- Line 134, please define OD, as well as add the ID, internal diameter.
- Lines 136 & 137: please check your internal (pipette) solution; to me it is hypoosmolar.
- Line 144,Please add "." after "(HEKA)".
- Lines 148-152, Sentence "TRPA1 inward/outward currents..." does not read well; please elaborate on what you have done; I suggest a schematic of current you have applied add as a panel to one of your figures.
- Line 175 please define "d6" as day 6.
- Line 188, define RQ as well as add it to your analysis section.
- Figure 3, question remains here is that if the current mediated is by ASIC channels, why the tmax is different between two groups?
- Figure 4 panel C, Have you tried on the naive cell line to see what you have seen earlier (in PH 6) is via TRPV1 or not?
- Line 273, Please define SE, I suggest to use SEM.
- Line 353, Please elaborate on why the time course is different.
Author Response
Thank you for your thorough review of our manuscript. We are grateful for the comments and criticism.
Major comments:
- The time course of activation of ASIC1a in the control and treated cells were different. If the same channel would be the responsible in both group; why there is the time course delay? Therefore, I believe to confirm this, the authors needs to repeat the experiments mentioned in figure 6 on the untreated cells. This way they can claim that the same or different channel is responsible for the findings observed in figure 4a.
Concerning the time course delay of ASICs on untreated and treated cells. We see 3 possible explanations for such a phenomenon: 1) the activation time can be affected by the slow acid application or solution exchange. We would like to kindly note that in our application system the solution change is very fast – less than 20 ms, according to the specifications of SF-77B Perfusion Fast Step (Warner instruments) which drives application tubes. The process is automatic and digitally controlled. The speed of solutions was constant and controlled by monitoring perfusion pressure. Also, the speed of application did not change throughout all the experiments in this work.
2) Such a delay can be evoked by low agonist concentration, which holds true for any channel. For example, we observed it while measuring pH dependence, i.e. at high pH (low H+). This is also not our case because the same pH 6.0 stimulus induced the currents shown on fig. 3.
3) The activation delay can be evoked by low channel responsiveness. Actually, the current amplitudes in untreated cells were extremely small (average 141 pA, n=14). We think that the delay in our case may be due to a very small abundance of functional ASIC channels which is also manifested in extremely low amplitudes. Contribution of slow channels might have been relevant, however, we show that it is not TRPV1 (please, see our response on comment 12 below).
All in all, since time delay of activation is a value that is not a kinetic constant and is extremely susceptible to different factors, we decided do not include this fact in our manuscript. The necessary corrections are made.
Concerning time delay on figure 4a, TRPV1 is a slowly-activating channel (unlike ASIC-channels) and its time of activation may vary significantly.
Concerning the fact that the same channels are responsible for currents in untreated and treated cells. After your request, we conducted a comparative test of the ligands’ action on undifferentiated SH-SY5Y cells (Appendix A, Fig. A4). Sevanol at the concentration of 300 μM suppressed acid-induced currents of untreated cells by 38.2 ± 3.3% (n = 5) which is 2 times less than the effect on currents from differentiated cells. Since sevanol does not act on ASIC2a isoform [10.1074/jbc.M112.366427] and the effect on heteromeric ASIC1a/ASIC2 has not been reported yet, we suppose that the currents from untreated cells are substantially mediated by channels composed either entirely of ASIC2 subunits or possibly containing it in heteromeric form. Peptide Mambalgin-2 at the concentration of 3 µM inhibited pH 6.0-induced currents by 49.2 ± 1.7% (n = 5) which is also significantly lower than the effect on the currents of differentiated cells. Mamb-2 was previously reported to act less effective on heteromeric ASIC1a/ASIC2a (IC50 = 246 nM) compared with homomeric ASIC1a (IC50 = 55 nM) [10.1038/nature11494, 10.1074/jbc.M114.561076]. Thus, the negative shift in Mamb-2 efficiency possibly confirms the greater contribution of ASIC2 subunits in mediating currents of undifferentiated cells. These results and discussion are added to the respective paragraphs.
- The authors mentioned that ASIC1a is involved in CAMKII maturation; it would be good that the authors compare the mRNA expression of CAMKII in the treated and untreated cells.
In literature, there are speculations on the possible coupled action of ASIC1a and CaMK II in neurodiffrerentiation. It is known that the direction of axon growth cones is guided by the joint switch-like action of calcineurin and CaMKII [10.1016/J.NEURON.2004.08.037]. The latter was also shown to affect the length of cytoskeletal proteins such as F‐actin [10.1002/jcp.28867]. Homomeric ASIC1a are permeable to Ca2+, and their activation was reported to be functionally coupled to the activation of CaMKII [10.1016/j.neuron.2005.10.011], therefore, the role of ASIC1a in neurodifferentiation could possibly be mediated by CaMKII signaling pathway.
After your request, we tried to compare the mRNA expression of CAMKII in the treated and untreated cells. Unfortunately, the primers for real time PCR produced significant background, and the attempts to optimize the reaction conditions were in vain.
However, even if we had managed to detect any significant change in CaMK II expression. we would hardly be able to make any substantial conclusion from it. CaMK II is regulated by Ca2+-calmodulin complex which induces autophosphorylation that renders the kinase active. So, mRNA expression would not provide information on the extent of active form. Also, we could not clearly say anything on where CaMKII localizes in cell or if it forms an active complex with ASIC1a during RA-treatment. All in all, the precise elucidation of CaMK II role would require different methods and specific experimental design. Such type of study is beyond the scope of our scientific interest.
Minor comments:
- line 22: remove "." after "acid-sensing ion channels.".
Corrected.
- In many places the authors have not defined the acronyms first such as in line 25, ASIC and TRP; please check the whole manuscript.
The mentioned acronyms were defined. The manuscript was further checked for any other undefined acronyms.
- Lines 44-47, the sentence "Human neuroblastoma..." does not read well; please revise.
We split this sentence in two: “Human neuroblastoma SH-SY5Y is a thrice cloned subline of the cell line SK-N-SH that was derived from bone marrow biopsy of a 4-year-old female patient. Since its establishment, this cell line has served as a tool for the study of various pathological conditions associated with neurodegenerative processes, such as Parkinson’s and Alzheimer’s diseases, amyotrophic lateral sclerosis, ischemia, as well as neurotoxicity and pathogenesis of viral infections”.
- Line 68, please remove "." after "clinical practice."
Corrected.
- Line 134, please define OD, as well as add the ID, internal diameter.
Defined and internal diameter is added.
- Lines 136 & 137: please check your internal (pipette) solution; to me it is hypoosmolar.
Osmolarity of pipette solution was adjusted with 1M sucrose. The difference in osmolarity between pipette and extracellular solutions was 10-15 mOsm/L. This information is added to the text.
- Line 144,Please add "." after "(HEKA)".
Added.
- Lines 148-152, Sentence "TRPA1 inward/outward currents..." does not read well; please elaborate on what you have done; I suggest a schematic of current you have applied add as a panel to one of your figures.
This sentence was modified. The scheme of the voltage protocol was added as Figure A1 in Appendix A and elaboration was given in the title to it.
- Line 175 please define "d6" as day 6.
Corrected.
- Line 188, define RQ as well as add it to your analysis section.
RQ (relative quantification) is defined in Methods, section 2.4. Quantitative PCR
- Figure 3, question remains here is that if the current mediated is by ASIC channels, why the tmax is different between two groups?
Fig. 3 is corrected. Please, see us addressing this issue above.
- Figure 4 panel C, Have you tried on the naive cell line to see what you have seen earlier (in PH 6) is via TRPV1 or not?
Yes. Consecutive application of capsaicin and pH 6.0 was done and it showed no response to capsaicin and did not diminish acid-induced currents any significantly compared with control (pre-application of pH 7.4). It suggests that TRPV1 rather does not contribute to acid-induced currents on untreated neuroblastoma. We give the corresponding traces in Appendix A, fig. A4.
- Line 273, Please define SE, I suggest to use SEM.
Corrected here and throughout the text. SEM was defined at the first mention.
- Line 353, Please elaborate on why the time course is different.
Corrections in the text are made. Please, see us addressing this issue above.
Reviewer 2 Report
In this paper, Kalinovskii and colleagues use undifferentiated human neuroblastoma SH-SY5Y cells and SH-SY5Y cells differentiated by treatment with retinoic acid (RA) to characterize endogenous acid-sensing ion channels (ASICs) and transient receptor potential (TRP) channels. Using qPCR, they found that ASIC1a is the main ASIC transcript in SH-SY5Y cells and that RA-differentiation increased its expression, while it sligthly decreased the expression of TRPV1 and TRPA1. Usind patch clamp electrophysiology, they found that the ASIC current amplitude was also increased by RA-differentiation and that TRPV1 and TRPA1 currents could not be elicited in RA-differentiated cells and only in a subset of undifferentiated cells. They go on to show that the ASIC current is mainly mediated by homomeric ASIC1a. By and large the experiments are carefully performed and the conclusions supported by the data. My main criticism is that this manuscript does provide only an incremental advance over what has been known previously. Almost 10 years ago, Xiong and colleagues already characterized ASIC currents in SH-SY5Y cells. Thus, the main advance of the current study is that differentiation by RA increases current amplitude approximately 3-fold. That is it. In my opinion this is not suffcient to warrant publication in “Biology”.
Minor comments:
1) The language would profit from some editing. Here are a few examples:
Line 72: “in comparison with its intact counterpart”, I think also differentiated cells are “intact”
Lines 174-175: “the number of transcripts was significantly higher on d6”, Higher than what?
Line 228-229: “cells fitted well with a one-exponential function”. It is not the cells but data that are fitted.
Lines 247-248: “TRPV1 and TRPA1 content”: what is meant here is the mRNA abundance; “content” is too unspecific.
2) Were the primers to detect ASIC1a and ASIC2a subtype-specific (that is not detecting 1b and 2b)? Please mention this explicitly.
3) Lines 182-183: “these data indicate that only ASIC1a may be reliably detected after 6 days of incubation”, it is difficult to predict current amplitudes just by the mRNA abundance. Therefore, I do not think that this conclusion can be drawn.
4) Lines 191-192: “Error bars are based on the Confidence level in the mean RQ Min/Max calculations.” This should be better explained.
5) Figure 2 and lines 198-200: from the figure it appears also untreated cells had “neuritic processes”. Moreover, it appears that cells with a round shape and without neurites have been selected for electrophysiological analysis. What is the significance then of the neuritic processes?
6) To my mind, it does not make sense to analyze the time to peak (line 222 and Fig. 3C). This will be unspecifically influenced by the speed of the solution exchange.
7) Lines 371-373: how can the action of APETx2 on voltage-gated sodium channels explain a reduction of proton-activated current amplitudes?
Author Response
Thank you for your thorough review of our manuscript. We are grateful for the comments and criticism.
In this paper, Kalinovskii and colleagues use undifferentiated human neuroblastoma SH-SY5Y cells and SH-SY5Y cells differentiated by treatment with retinoic acid (RA) to characterize endogenous acid-sensing ion channels (ASICs) and transient receptor potential (TRP) channels. Using qPCR, they found that ASIC1a is the main ASIC transcript in SH-SY5Y cells and that RA-differentiation increased its expression, while it sligthly decreased the expression of TRPV1 and TRPA1. Usind patch clamp electrophysiology, they found that the ASIC current amplitude was also increased by RA-differentiation and that TRPV1 and TRPA1 currents could not be elicited in RA-differentiated cells and only in a subset of undifferentiated cells. They go on to show that the ASIC current is mainly mediated by homomeric ASIC1a. By and large the experiments are carefully performed and the conclusions supported by the data. My main criticism is that this manuscript does provide only an incremental advance over what has been known previously. Almost 10 years ago, Xiong and colleagues already characterized ASIC currents in SH-SY5Y cells. Thus, the main advance of the current study is that differentiation by RA increases current amplitude approximately 3-fold. That is it. In my opinion this is not suffcient to warrant publication in “Biology”.
We are thankful for your review and criticism on our work and we are sorry that you cannot recommend it for publication. In our defense, we would like to gently note that the study done by Xiong et al. concerned aspects of vesicular release upregulated by acidification in undifferentiated neuroblastoma. As the authors themselves point out in the abstract “[their] results provide a preliminary study on ASICs in SH-SY5Y cells and neurotransmitter release”. To our knowledge, no further publications on this topic were made. The aim of our work was to demonstrate that retinoic acid-treated SH-SY5Y cells are an effective model for the pharmacological study of native human ASIC1a channels, which have undoubted physiological and pharmacological significance. According to our data, which were added to this manuscript (see Appendix A, Fig. A4), as well as according to the data presented by Xiong et al., acid-induced currents from undifferentiated cells are mediated by a heterogeneous mixture of channels, which makes these cells much less convenient as a model.
Below we address the minor comments:
1) The language would profit from some editing. Here are a few examples:
Line 72: “in comparison with its intact counterpart”, I think also differentiated cells are “intact”
Changed for “untreated” or “undifferentiated” here and throughout the text.
Lines 174-175: “the number of transcripts was significantly higher on d6”, Higher than what?
Corrected for “the number of mRNA transcripts was significantly higher on day 6 of RA-treatment compared with control (untreated cells)”.
Line 228-229: “cells fitted well with a one-exponential function”. It is not the cells but data that are fitted.
The full sentence was “Acid-induced currents of untreated and RA-treated SH-SY5Y cells fitted well with a one-exponential function”.
Lines 247-248: “TRPV1 and TRPA1 content”: what is meant here is the mRNA abundance; “content” is too unspecific.
Changed for “mRNA abundance of TRPV1 and TRPA1”.
2) Were the primers to detect ASIC1a and ASIC2a subtype-specific (that is not detecting 1b and 2b)? Please mention this explicitly.
ASIC2 primers detect both isoforms. We changed primers for ASIC1a and now they are specific to ASIC1a. We made additional experiments on quantification of ASIC1a transcript (transcript variant 2 [DOI 10.1074/jbc.M110.171330]). Data are shown on figure 1. We added to text information on primers specificity.
3) Lines 182-183: “these data indicate that only ASIC1a may be reliably detected after 6 days of incubation”, it is difficult to predict current amplitudes just by the mRNA abundance. Therefore, I do not think that this conclusion can be drawn.
We agree on this point. Neither mRNA nor protein amounts can be absolutely reliable predictive marker of the functional channel quantity on the membrane. The sentence is removed.
4) Lines 191-192: “Error bars are based on the Confidence level in the mean RQ Min/Max calculations.” This should be better explained.
We recalculated data to traditional mean±SEM.
5) Figure 2 and lines 198-200: from the figure it appears also untreated cells had “neuritic processes”. Moreover, it appears that cells with a round shape and without neurites have been selected for electrophysiological analysis. What is the significance then of the neuritic processes?
Undifferentiated cells indeed bear neurites, that are, however, shorter and less branched than those of RA-treated cells. In general, undifferentiated cells show traits more or less characteristic to tumour cells. RA-treatment results in longer neurites, dendrite extension and branching, growth cones, and formation of apparent “intercellular contacts”. This appearance is more intrinsic to neuronal-like. The cells for patch clamping on fig. 2 (black arrows), in fact, have processes: the middle cell has a well-developed neurite with terminal branching. The upper and lower cells also have branched dendrites. The round shape is also significant because it distinguishes N-type cells from flat angular S-type cells that are not eligible for patch clamping (photo and representative whole-cell current in Appendix A, fig A3).
6) To my mind, it does not make sense to analyze the time to peak (line 222 and Fig. 3C). This will be unspecifically influenced by the speed of the solution exchange.
We have to agree with you on this point. Necessary corrections are made- panel 3C and discussion deleted. However, we would like to kindly note that solution exchange in our system is very fast. The system is driven by SF-77B Perfusion Fast Step (Warner instruments) and is digitally controlled. The speed of the solution exchange is determined by the step speed which is 20 ms for 700 µm step, according to the device specifications. The flow speed of solutions was also fixed at a constant value for all measurements and monitored via the controller of the perfusion pressure.
7) Lines 371-373: how can the action of APETx2 on voltage-gated sodium channels explain a reduction of proton-activated current amplitudes?
We agree that it is irrelevant in this context. Reduction of currents is apparently due to nonspecific action of APETx2 on ASIC1a at high concentrations, as it was shown before in doi:10.1111/bph.14089. The respective sentence in the Discussion is corrected.
Reviewer 3 Report
The authors in this study investigated the expression and function of acid-sensing ion channels in differentiated neuroblastoma cells (SH-SY5Y) treated by retinoic acid. They found that retinoic acid significantly increased the expression of ASIC1a but not ASIC2. The currents from the differentiated cells are suggested to be elicited by the homomeric ASIC1a channel. In addition, retinoic acid reduced the expression of TRPV1 and TRPA1 channels. The manuscript is well-written, and the methods are appropriate. I only have a few concerns below.
1 Is it possible to show the raw RNA bands for figure 1?
2 Have the authors tested the protein expression levels of these channels in differentiated neuroblastoma cells (SH-SY5Y) treated by retinoic acid?
3 Line 241, please provide the reference of the TRPA1 selective agonist.
4 If possible, please show the IC50 for capsaicin and ASP7663, respectively. It is possible that TRPV1 and TRPA1 need a bigger concentration of agonists to be activated. Also, any selective antagonists for these two channels?
Author Response
Reviewer3
Thank you for reviewing our manuscript, we are grateful for your comments and concerns. Below are our responses.
1 Is it possible to show the raw RNA bands for figure 1?
We added electropherogram to AppendixA Figure A2. But we should note that some components of the buffer for real-time PCR make DNA bands diffused.
2 Have the authors tested the protein expression levels of these channels in differentiated neuroblastoma cells (SH-SY5Y) treated by retinoic acid?
Protein expression was not tested. We think functional testing is the better way to verify the quantity of functional channels in the cell membrane.
3 Line 241, please provide the reference of the TRPA1 selective agonist.
The reference was added.
4 If possible, please show the IC50 for capsaicin and ASP7663, respectively. It is possible that TRPV1 and TRPA1 need a bigger concentration of agonists to be activated. Also, any selective antagonists for these two channels?
EC50s and respective references were added to the text.
Concerning TRPV1, the EC50 of capsaicin depends on a variety of factors and varies from 110 nM to 1.11 µM for human TRPV1 (https://doi.org/10.1016/S0896-6273(00)80564-4, DOI:10.1523/JNEUROSCI.4691-05.2006). 1 µM of capsaicin is typically used in electrophysiology on heterologously expressed TRPV1 (https://doi.org/10.1016/S0896-6273(00)80564-4). We also took an excessive 4 µM of capsaicin since 1µM did not show detectable responses.
Concerning TRPA1, 0.7 µM of ASP7663 was taken for tests (EC50 = 0.51 µM for human TRPA1). This ion channel is known for agonist-induced desensitization so we took the concentration slightly above EC50 and did not use excessive concentration of agonist here.
The currents on untreated neuroblastoma were so scarce that we could not perform any tests with antagonists. RA-treatment reduced TRPV1 and TRPA1 mRNA levels and we could not detect any currents at all.
Round 2
Reviewer 1 Report
No further comment
Author Response
We thank the reviewer for the positive evaluation of the revised version of the manuscript.
Reviewer 2 Report
The authors have improved their mansucript. I have one remaining comment:
Line 436-437: the authors speculate that currents from untreated cells are mediated by channels composed entirely of ASIC2a subunits (ASIC2a homomer). But the pH used in the experiment reported in Figure A5, pH 6, will not activate the ASIC2a homomer, excluding a contribution by this channel to the currents that were analysed.
Author Response
We would like to thank you for the constructive comments.
Line 436-437: the authors speculate that currents from untreated cells are mediated by channels composed entirely of ASIC2a subunits (ASIC2a homomer). But the pH used in the experiment reported in Figure A5, pH 6, will not activate the ASIC2a homomer, excluding a contribution by this channel to the currents that were analysed.
We agree with you. We changed the sentence.
line 434. Since sevanol does not act on ASIC2a isoform [33], the effect on heteromeric ASIC1a/ASIC2 has not been reported yet, and pH 6.0 does not activate homomeric ASIC2a channels [28], we suppose that the currents from untreated cells are partially mediated by channels containing ASIC2 subunits in heteromeric form.
Reviewer 3 Report
The authors have addressed all my concerns.
Author Response
We thank the reviewer for the critical evaluation of the manuscript, which we think has helped to improve the manuscript.